# Visual Navigation Using Inverse Reinforcement Learning and an Extreme Learning Machine

**Qiang Fang \*,†, Wenzhuo Zhang † and Xitong Wang †**

College of Intelligence Science and Technology, National University of Defense Technology, Changsha 410073, China; doublezore00@163.com (W.Z.); Xitongwang2021@163.com (X.W.)
\* Correspondence: qiangfang@nudt.edu.cn
† These authors contributed equally to this work.

**Abstract:** In this paper, we focus on the challenges of training efficiency, the designation of reward functions, and generalization in reinforcement learning for visual navigation and propose a regularized extreme learning machine-based inverse reinforcement learning approach (RELM-IRL) to improve the navigation performance. Our contributions are mainly three-fold: First, a framework combining extreme learning machine with inverse reinforcement learning is presented. This framework can improve the sample efficiency and obtain the reward function directly from the image information observed by the agent and improve the generation for the new target and the new environment. Second, the extreme learning machine is regularized by multi-response sparse regression and the leave-one-out method, which can further improve the generalization ability. Simulation experiments in the AI-THOR environment showed that the proposed approach outperformed previous end-to-end approaches, thus, demonstrating the effectiveness and efficiency of our approach.

**Keywords:** visual navigation; inverse reinforcement learning (IRL); extreme learning machine (ELM); deep learning; A3C





## 1. Introduction

With the rapid development of artificial intelligence technologies, the research and application of unmanned platforms, such as robots, has become a research hotspot. The question of how to endow machines with higher autonomy and adaptability is an important goal in current research. The navigation of mobile robots has always been one of the most basic tasks in the field of robotics [1–4]. The classic navigation control method used for robots is based on the establishment of an environment map, positioning using sensor (such as GPS and IMU) information, and finally path planning and tracking [5,6].

In recent years, with the rise of deep learning (DL), research on using DL to achieve end-to-end autonomous navigation of mobile robots has become a popular topic. However, navigation methods based on DL not only require the robot to accurately position itself but also require a large amount of human prior knowledge. This is not in line with the human way of thinking. Therefore, the question of how to give robots self-learning and self-adaptive capabilities in navigation control, as well as how to teach them to use visual information to control their behavior, is a problem worth studying.

Reinforcement learning (RL) relies on interaction and feedback between agents and the environment, prompting agents to constantly update their own strategies and thereby improving their ability to make decisions and control their own behavior [7–9]. The visual navigation method based on RL does not require knowledge of the accurate real-time position of the robot but can realize end-to-end action outputs based on inputting the current observation image. This method is similar to the way of human thinking—using continuous trial and error training in the environment to adjust the strategy and, thereby, improving the navigation control ability.

However, RL is currently facing many problems. The use of constant trial and error causes a low RL data utilization rate, requires a great deal of training time, makes it difficult to properly set the reward function for different tasks, and causes the adaptability and transfer ability of scene changes to be poor. Imitation learning makes use of expert data to provide certain guidance for agents, thereby, improving the training efficiency and adaptability of RL. The common methods of imitation learning include supervised regression [10] (such as behavioral cloning [11]), inverse reinforcement learning (IRL) [12], and direct strategy interaction [13]. IRL makes use of the reward function from the expert trajectory or expert strategy to guide the update of the strategy.

In previous research, mobile robot navigation based on IRL has been realized. This type of navigation can obtain the reward function from the expert trajectory and provide more accurate feedback for navigation control strategies for robot training, thereby, speeding up the training speed of the algorithm, improving the accuracy of navigation control, and making the navigation path closer to the optimal path [14–16]. However, IRL still has certain limitations.

The improvement of the navigation performance is based on the reward function obtained from the target point corresponding to the expert trajectory. For target points without expert trajectories, the navigation effect of the algorithm will be reduced, which means that the generalization ability of the model will not be strong. For a robot navigation algorithm that can be applied in reality, in addition to accurate navigation and control capabilities for trained targets, it also needs to have good migration and generalization capabilities for unknown environments and tasks. Therefore, designing an algorithm that improves the generalization performance while maintaining navigation control accuracy would be an important research development.

To solve this problem, this paper proposes an IRL navigation control method based on a regularized extreme learning machine (RELM-IRL). First, the IRL is used to calculate the reward function from the expert trajectory. Then, the ELM is used to perform regression analysis on the reward function of the expert trajectory in order to obtain a more accurate reward function. Finally, the regularized ELM further improves the generalization ability of the model. The experimental results show that, compared with the traditional RL and IRL, the IRL navigation control method based on a regularized ELM had better generalization and migration capabilities and could navigate to the target point more accurately.

We highlight the main novelty and contribution of this work compared with existing mobile robot navigation methods as follows:

- A framework combining extreme learning machine with inverse reinforcement learning is presented. This framework obtains the reward function directly from the image information observed by the agent and improves the generation to the new environment. Moreover, the extreme learning machine is regularized by multi-response sparse regression (MRSR) and leave-one-out (LOO), which can improve further improve the generalization ability.
- Simulation experiments show that the proposed approach outperformed previous end-to-end approaches, which demonstrates the effectiveness and efficiency of our approach.

The remainder of the paper is organized as follows: we introduce the related work in Section 2 and review the research background in Section 3. Our approach is proposed in Section 4. Experiments in the AI2-THOR environment are evaluated, and the results are discussed in Section 5. Finally, our conclusions are drawn in Section 6.

## 2. Related Work

IRL was originally proposed by Ng.A.Y and Russell, who discussed three aspects: IRL in finite state space, IRL in infinite state space, and IRL sampling trajectory [17]. In 2004, Abbeel and Ng.A.Y proposed a new method of IRL—apprenticeship learning [18]. In this method, the reward function is established through the expert example, so that the optimal strategy obtained by the reward function is close to the expert example strategy. The maximum marginal programming (MMP) algorithm is similar to apprenticeship learning

using linearized rewards in that it finds a strategy that makes the expert trajectory under this strategy better than the other generated trajectory and minimizes the cost function between observation and prediction [19].

Ratliff proposed the LEARCH algorithm by extending the maximum marginal planning and applied it to outdoor autonomous navigation. Bayesian IRL uses a Bayesian probability distribution to deal with ill-posed problems [20]. It samples the state-behavior sequence from the prior distribution of the possible reward function and calculates the posterior of the reward function using Bayes. Similar to Bayesian IRL, maximum entropy IRL uses a Markov decision model to calculate the probability distribution of state behavior [21]. It focuses on the distribution of trajectories rather than simple actions.

Zhu [22] proposed a target-driven model based on an actor–critic structure for mobile robot navigation control. It takes the observation image of the camera as the input of the model and the control instructions of the agent as the output, thus, realizing end-to-end navigation indoors. At the same time, the target image and the currently acquired image are combined to improve the generalization ability of the model. In 2019, Mitchell Wortsman designed an improved navigation model using meta-learning and self-supervised learning based on the "object-driven model" [23]. It can update the network model by providing additional gradients for itself through its internal self-supervised network, which greatly improves the performance of the model and enhances the generalization.

In addition, some researchers have used depth images as inputs to train agents. For example, Li K used depth images to realize the robot's navigation and obstacle avoidance function [24]. In terms of images, some researchers combined the attention mechanism and image segmentation technology to provide additional navigation information for the agent. For example, Keyu Li and Yangxin Xu improved the attention RL algorithm by introducing a dynamic local goal setting mechanism and a map-based safe action space to achieve navigation in an indoor environment [25].

Researchers have explored many ways to improve the generalization performance of mobile robot navigation control. K Cobbe trained an agent using multiple environments with different appearances and layouts, hoping to allow the agent to learn a wider range of strategies by training it on as many environmental tasks as possible [26]. Experiments showed that, as the number of Markov decision processes used for the training increased, the success rate of the agent in unknown new tasks gradually increased. This shows that the generalization performance of the model was improved.

K Lee used a clever image enhancement method to improve the generalization of a model. By inserting a random convolutional layer between the environmental input image and the policy decision, the input image could be randomly converted into the expression of different features before each batch of training starts. Different but similar inputs can help the model understand subtle changes in the environment and learn more general strategies. In addition, using meta-learning [27] and transfer learning [28,29] to improve the generalization performance of the model is also an important research direction.

## 3. Background

In this section, we will provide some background to our chosen approach. Since our approach is based on a combination of inverse reinforcement learning (IRL) and extreme learning machines (ELMs), we will briefly introduce the IRL first and then detail the ELM.

### 3.1. Inverse Reinforcement Learning (IRL)

Inverse reinforcement learning involves learning the reward function from expert data. We assume that when experts complete a task, their decisions are usually optimal or nearly optimal. When the cumulative reward function expectations generated by all the strategies are not greater than the cumulative reward function expectations generated by the expert strategy, the reward function of RL will be the reward function learned from the expert data. Apprenticeship learning [30,31] is a type of IRL, which sets the prior basis function as the reward function. This ensures that the optimal strategy obtained from the

reward function is near the expert strategy using the given expert data. We suppose the unknown reward function:

$$R(s) = w * \phi(s) \tag{1}$$

where $\phi(s)$ is the basis function. At this time, the IRL requires the coefficient $w$. This is defined by the value function:

$$V^\pi(s) = R^t(s) + \gamma R^{t+1}(s) + \gamma^2 R^{t+2}(s) + \dots$$
$$= \sum_{t=0}^{\infty} \gamma^t R^t(s) \tag{2}$$

Combining Equations (1) and (2), we can obtain:

$$V^\pi(s) = \sum_{t=0}^{\infty} \gamma^t \sum_{i=1}^{d} w_i \phi(s_t)$$
$$= \sum_{i=1}^{d} \sum_{t=0}^{\infty} \gamma^t w_i \phi(s_t)$$
$$= \sum_{i=1}^{d} w_i \sum_{t=0}^{\infty} \gamma^t \phi(s_t) \tag{3}$$
$$= \sum_{i=1}^{d} w_i V_i^\pi(s)$$

Among these, $V_i^\pi$ represents the cumulative value of a certain dimension mapping feature, and $d$ represents the number of dimensions of $w$.

According to the definition of the optimal strategy, we know that, in each state, the value function of the optimal strategy $\pi^*$ should not be weaker than the value function of other strategies. This means that:

$$V_{\pi^*}(s) \geq V_\pi(s) \tag{4}$$

Our ultimate goal is to find the best reward function to maximize the value of the optimal strategy:

$$E_{s' \sim P_{s,\pi^*(s)}}[V^\pi(s')] \geq E_{s' \sim P_{s,a}}[V^\pi(s')] \tag{5}$$

$$\sum_{i=1}^{d} w_i \left( E_{s' \sim P_{s,\pi^*(s)}}[V^\pi(s')] - E_{s' \sim P_{s,a}}[V^\pi(s')] \right) \geq 0 \tag{6}$$

where $P_{s,a}$ represents the strategy of taking action a at state s. In addition, we add two constraints to the objective function:

(1) We limit the size of the reward function. For the linear programming problem, we set a restriction on the reward value:

$$|R| \leq R_{\max} \tag{7}$$

(2) We only consider the differences between the optimal strategy and a sub-optimal strategy. Considering the diversity of strategies and the influence of other strategies on the calculation error, we only consider the gap between the optimal strategy and the sub-optimal strategy:

$$\max \sum_{s \in S_0} \min_{a \in \{a2,\dots ak\}}$$
$$\left\{ \sum_{i=1}^{d} w_i \left( E_{s' \sim P_{s,\pi^*(s)}}[V^\pi(s')] - E_{s' \sim P_{s,a}}[V^\pi(s')] \right) \right\} \tag{8}$$

Among these, we first find the sub-optimal strategy closest to the optimal strategy and then obtain the reward function by maximizing their gap.

### 3.2. Extreme Learning Machine (ELM)

The extreme learning machine (ELM) was proposed by Professor Huang Guangbin from Nanyang Technological University in 2004 [32]. As shown in the figure, it is the same as a single hidden layer feedforward neural network consisting of an input layer, a hidden layer, and an output layer. However, an innovative point and advantage of the ELM is that it randomly initializes and fixes the weight vector $W$ and the bias vector $b$ between the input layer and the hidden layer.

It does not need to be updated according to the gradient in the subsequent training, which greatly reduces the amount of calculation needed. Secondly, it can directly determine the weight vector $\beta$ between the hiddELMen layer and the output layer through the method of least squares or inverse matrix, without iterative update, thus, speeding up the calculation. Research shows that, under the condition of ensuring accuracy, the learning speed and generalization performance of the ELM are better than those of the traditional neural network.

Figure 1 is an ELM with $L$ hidden neurons. There are $N$ samples $\{X_i, T_i\}$ in the training set, where $X_i = [x_{i1}, x_{i2} \cdots x_{in}]^T$ is the input and $T_i = [t_{i1}, t_{i2} \cdots t_{in}]^T$ is the output—that is, the label. For the input Xi, the function output after the hidden layer can be expressed as:

$$h_i(x) = g(w_i \cdot x_i + b_i) \tag{9}$$

where $w$ is the weight, representing the strength of the connection between neurons, and $b$ is the bias, with $w$ and $b$ being randomly initialized and fixed before the network training. $g(x)$ is the activation function, and $h_i(x)$ is the output of $i$-th hidden layer neurons. The final output can be expressed as:

$$o_j = \sum_{i=1}^{L} \beta_i g(w_i \cdot x_j + b_i) j = 1, \ldots N \tag{10}$$

The final expected result of the neural network is the minimum error between the output and the label, which is:

$$\sum_{j=1}^{N} \left\| o_j - t_j \right\| \tag{11}$$

This means that there are $\beta_i$, $w_i$, and $b_i$, such that:

$$\sum_{i=1}^{L} \beta_i g(w_i \cdot x_j + b_i) = t_j, j = 1, \ldots N \tag{12}$$

This is expressed as a matrix:

$$H\beta = T \tag{13}$$

where $\beta$ is the weight of the output layer and $T$ is the expected output, which is the label provided by the data set. $H$ is the output of the hidden layer, which can be expressed as:

$$H = \begin{bmatrix} g(w_1 \cdot x_1 + b_1) \cdots g(w_L \cdot x_1 + b_L) \\ \ddots \\ g(w_1 \cdot x_N + b_1) \cdots g(w_L \cdot x_N + b_L) \end{bmatrix}_{N*L} \tag{14}$$

The output weight and the desired output can be expressed as:

$$\beta = [\beta_1^T, \ldots, \beta_L^T]^T, T = [T_1^T, \ldots, T_N^T]^T \tag{15}$$

Traditional gradient descent neural networks usually iterate the network parameters $w$ and $\beta$ by minimizing the loss function as follows:

$$\epsilon = \sum_{j=1}^{N} \left( \sum_{i=1}^{L} \beta_i g(w_i x_j + b_i) - t_j \right)^2 \tag{16}$$

In the ELM, the values of $w$ and $b$ are fixed, meaning that the value of $H$ is also uniquely determined. Therefore, the optimal $\beta$ value can be obtained by inverting the matrix:

$$\hat{\beta} = HT \tag{17}$$

where $H'$ is the generalized inverse of matrix $H$, and it has been proven that the solution is unique. Compared with the ordinary single hidden layer feedforward neural network, ELM guarantees the accuracy, reduces the amount of calculation by fixing the parameters of the input layer and enables $\beta$ to be directly obtained in a linear manner. This greatly improves the calculation speed, while also improving the generalization performance of the model.

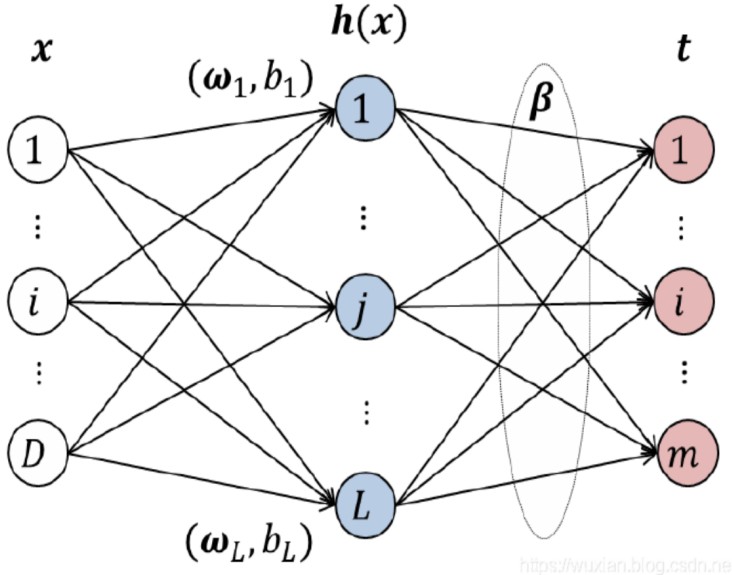

**Figure 1.** Diagram of the extreme learning machine structure.

## 4. Our Approach

In this section, the design of a navigation algorithm based on IRL and regularized ELM to improve the navigation performance of a mobile robot is described. As shown in Figure 2, the algorithm structure we designed consists of three parts: the first part is a general feature extraction part, which uses resnet-50 to extract the features of the training scene; the second part is based on regularized ELM-IRL, using regularized ELM-IRL learning expert data rewards, as part of the input of the RL; the third part is an Asynchronous Advantage Actor-Critic (A3C) network [33]. It uses the fusion features of the fully connected layer and the reward function calculated by the regularized ELM to train the agent's action strategy.

During training, we first train the IRL part of the algorithm. The reward function is calculated by sampling several expert trajectories in different scenarios. On the other hand, the environment provides the current state image and the target image observed by the agent. These are mixed through the feature extraction network to obtain the state vector. Each time the agent takes an action, the parameters solved by the IRL and the features of the fully connected layer provide reward feedback together as the update source of the policy network and the value network parameters.

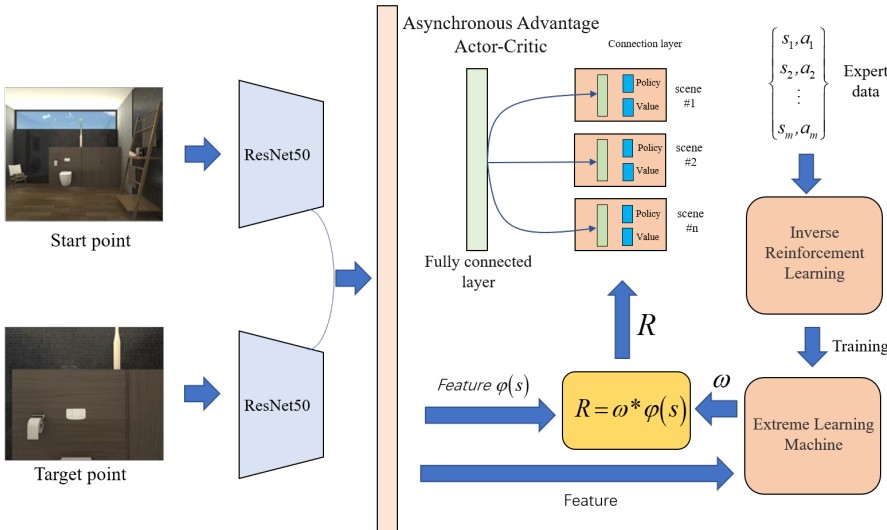

**Figure 2.** Framework diagram of our algorithm (RELM-IRL). First, resnet-50 is used to extract the features of the observed image. Meanwhile, the ELM is trained with the features and reward function found by IRL. Then, we regularize the ELM and finally input the obtained reward function to obtain the navigation strategy.

### 4.1. Mobile Robot Navigation Control Based on IRL

After IRL has obtained the reward function parameters $w$ calculated by the expert trajectories of different target points, the RL based on A3C can begin to train the agent action strategy. The A3C algorithm is based on the actor–critic structure and adopts asynchronous multi-threading. At the beginning, multiple training threads are created at the same time. Each training thread corresponds to a task. At the same time, it initializes its own network parameters and assigns them to each sub-thread. After that, all sub-threads are turned on in parallel and the gradient is accumulated in different tasks and environments. When the sub-threads themselves are updated, the gradient accumulation of all the sub-threads is used to update the A3C total parameters.

After a batch of iterations, the total parameters are again assigned to the sub-threads and a new round of iterative training is started. Since the threads are independent of each other and all threads update the global network together, this method can eliminate inter-sample correlation and improve the training speed. Each thread corresponds to a training task—that is, it navigates to a certain target point in the scene. During training, the input is the current state image and target image observed by the camera.

After the feature extraction part of the network, the mixed state feature is obtained. Then, the value function network and the strategy network generate value estimation and action prediction. After the execution of the action, according to the weight value of w obtained in the IRL part combined with the current input state feature vector to obtain the reward value and feedback to the network, the value network and policy network parameters can be updated. After a certain number of iterations have occurred, the network will reach convergence. The overall algorithm flow is as shown in Algorithm 1.

For each state of the agent, its state value function can be expressed as:

$$V(s_n) = r_0 + \gamma r_1 + \gamma^2 r_2 + \ldots \gamma^n r_n \tag{18}$$

For any action made by the agent, there is also a state–action value function to evaluate the pros and cons of the action:

$$Q(s, a) = r + \gamma V(s') \tag{19}$$

In training, the value function of each step is estimated by the value network, and there is no need to wait for the end of a test to calculate the value function. This means that the network can be updated in one step, which improves the efficiency. The error of calculating td is:

$$\delta(t) = Q(s,a) - V(s) = r + \gamma V(s') - V(s) \tag{20}$$

The policy network update formula is:

$$\theta = \theta + \alpha \nabla_\theta log \pi_\theta(s_t, a_t)\delta(t) \tag{21}$$

The value network update formula is:

$$\theta_v = \theta_v + \partial \delta^2 / \partial \theta_v \tag{22}$$

---

**Algorithm 1** Robot navigation algorithm based on IRL

---

**Initialization parameters:** $\theta, \theta_v$: Global network strategy function, value function parameter

$\theta', \theta'_v$:Threaded network strategy function, value function parameter

$t \leftarrow 1$

1: **repeat**
2:    $d\theta, d\theta_v, d\theta', d\theta'_v = 0$
3:    $\theta' \leftarrow \theta, \theta'_v \leftarrow \theta_v$
4:    $t_{start} = t$
5:    Random initialization starting point $s_t$
6:    **repeat**
7:      Select action $a$ according to strategy function $\pi(a_t|s_t, \theta')$;
8:      Execute $a$ to get the next state $s_{t+1}$;
9:      $t \leftarrow t + 1$;
10:     $T \leftarrow T + 1$;
11:    **until** terminal $s_t$ or $t - t_{start} = t_{\max}$
12:    $R = \begin{cases} 0, & \text{Suspension state} \\ V(s_t, \theta'_v), & Else \end{cases}$
13:    **for** i$\in \{t - 1, \ldots, t_{start}\}$ **do**
14:      Calculate the reward value $r$ of the environment feedback based on $r = w * \phi(s)$;
15:      $R \leftarrow r_i + \gamma R$;
16:      Update value function parameters:
       $\theta_v = \theta_v + \partial \delta^2 / \partial \theta$
17:      Update strategy function parameters:
       $\theta = \theta + \alpha \nabla_\theta log \pi_\theta(s_t, a_t)\delta(t)$
18:    **end for**
19: **until** $T > T_{\max}$

---

### 4.2. Reward Function Design Combining ELM and IRL

In order to solve the limitations of the IRL-based robot navigation algorithm in new tasks and new scenarios, it is necessary to improve the generalization performance of the algorithm. We used the state vector obtained from the current image and the target image to train the ELM, meaning that the reward function can be adjusted in real time according to the current state and the change in the target image during training, thus guiding the agent to navigate to the target point.

Traditional neural networks based on gradient descent usually update the network parameters by minimizing the loss function.

$$\varepsilon = \sum_{j=1}^{N} \left( \sum_{i=1}^{L} \beta_i g(w_i \cdot x_j + b_i) - t_j \right)^2 \tag{23}$$

where $w$ is the weight, which represents the strength of the connection between neurons; $b$ is the bias; and $\beta$ is the weight of the output layer. In the ELM, since the values of $w$ and $b$ are fixed, the output $H$ of the hidden layer is also uniquely determined. Therefore, the optimal $\beta$ value can be obtained by inverting the matrix:

$$\hat{\beta} = H'T \tag{24}$$

Here, $H'$ is the generalized inverse of matrix $H$. It has been proven that the $\hat{\beta}$ obtained by solving is unique. Compared with the ordinary single hidden layer feed-forward neural network, the ELM not only guarantees the accuracy, but also reduces the amount of calculation by fixing the parameters of the input layer and enables $\beta$ to be directly obtained in a linear manner. This greatly improves the calculation speed, while also improving the generalization performance of the model.

Although each expert trajectory is the optimal path from the start point to the end point, each trajectory has its own characteristics. The parameters of the reward function are different for different trajectories. In order to obtain a network that can better calculate the reward function through ELM training, it is necessary to analyze the part of the expert trajectory that has the greatest impact on the performance of the model. However, the parameters obtained from the expert trajectories of different target points only greatly improve their own navigation effect.

For trajectories with the same target point but different starting points, the obtained parameters improve the performance of the model, and the gap between them is very small. Considering the diversity of the intermediate process of the trajectory, we use the starting point and target point of the trajectory as the input of the ELM and the corresponding reward function parameters as the expected output of the ELM. The ELM is trained by sampling multiple expert trajectories with different starting points and target points, meaning that it can more accurately predict the reward function parameters of the new starting point and target point as the feedback of the model. Part of the network structure is shown in Figure 3, for each training target point several expert trajectories with random starting points are collected, including all state points $s$ and actions $a$ on the optimal path from the random starting point to the target point, which are recorded as an action–state pair $\{S_{random}, a_0; S_1, a_1; \cdots\cdots S_{t\,\arg et1}, a_n\}$. We then use the IRL to solve the corresponding reward function parameter $W$, where $W$ is a high-dimensional vector that is denoted as $\{w_1, w_2 \cdots\cdots w_n\}_n$.

We extract the image features of the start and end points of each trajectory in the expert trajectory as the input data of the ELM and the corresponding reward function parameter $w$ as the expected output. According to the ELM training method, we train the ELM to obtain the final parameter $\beta$. After this, the obtained ELM model is saved and, thus, the reward function parameter prediction network model combining the ELM and IRL is obtained. Through the trained ELM model, the starting point and target point of each task can be predicted. A more accurate expert reward function is designed to expand the scope of application of the previous model while at the same time improving the generalization performance of the model.

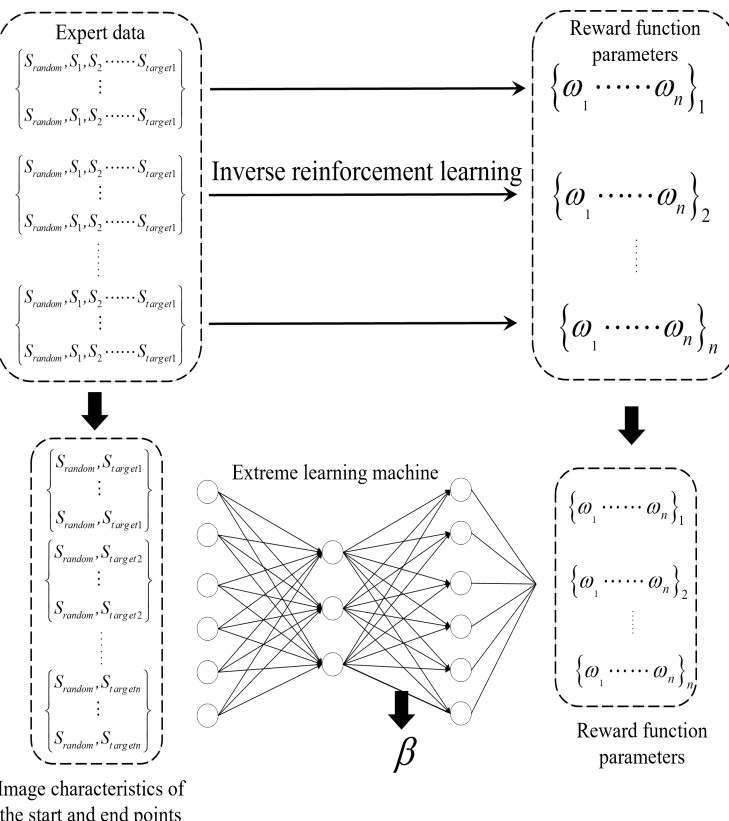

**Figure 3.** Structure diagram of extreme learning machine combined with expert trajectory.

### 4.3. Regularized ELM

The ELM is trained with expert data, enabling the model to predict the reward function parameter w according to the current state and the target point state, imitating the process of IRL to solve the parameters from the expert trajectory. However, as mentioned in the previous chapter, due to the high dimensionality of the data that need to be fitted, a large number of expert trajectories are needed to ensure the accuracy of the ELM predictions, which hinders the promotion and application of the model. On the other hand, in the face of large-scale data, ELM has the defect of a weak robustness and is easily affected by some interference data, which will cause large errors in the regression and make the model more complicated.

Therefore, we propose to use an algorithm based on regularized ELM to obtain a model with fewer parameters that is more concise and has a stronger generalization performance. As shown in Figure 4, the ELM model can be optimized by the regularization method. First, we complete the training of the ELM. After the training is completed, the MRSR is used to sort the hidden layer neurons. Afterward, in order to determine the optimal number of neurons to retain, LOO is used to verify the optimal model structure, then finally the optimal model structure and the trained model are obtained.

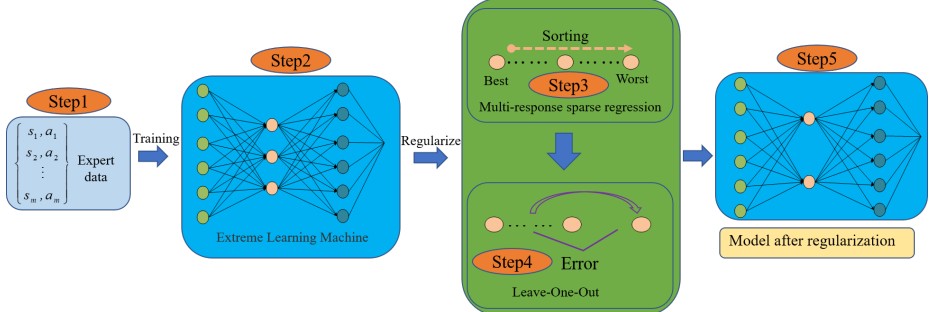

**Figure 4.** Algorithm structure of rgw regularized extreme learning machine.

### 4.3.1. Multi-Response Sparse Regression to Screen Hidden Layer Neurons of ELM

In order to reduce the complexity of the model and improve the generalization performance of the model, the method of multi-response sparse regression (MRSR) is used to sort the hidden layer neurons and the sorting criterion is the degree of influence of neuron output. Suppose a target variable $T = \begin{bmatrix} t_1 \dots t_p \end{bmatrix}$; the matrix size is $n * p$; and the input data $X = [x_1 \dots x_m]$ are a matrix of $n * m$—that is, the regression variable. The idea of MRSR is to use the following formula to cause $Y$ to tend toward $T$:

$$Y^k = XW^k \tag{25}$$

where $W^k$ is the weight coefficient matrix of the $k$-th step, with $k$ non-zero row vectors in the $k$-th step, while $Y^k = \begin{bmatrix} y_1^k \dots y_p^k \end{bmatrix}$ is the target variable estimated by $W$ and the regression variable $X$. At each step, MRSR will replace a zero row vector of $W$ with a non-zero row vector through calculation, which is equivalent to selecting one of the regression variables $X$ and adding it to the model. When $Y$ is a row vector—that is, when the model has a single output—the multi-response sparse regression degenerates to the smallest angle regression (LARS), meaning that MRSR can be seen as an extension of LARS.

Initially, $k = 0$, $Y^0$ and $W^0$ are set to zero matrices, and we normalize $X$ and $T$. At this time, we define the cumulative correlation between the $j$-th regressor $x_j$ and the current residual:

$$c_j^k = \left\| \left( T - Y^k \right)^T x_j \right\|_1 = \sum_{i=1}^{p} \left| \left( t_i - y_i^k \right)^T x_j \right| \tag{26}$$

When it comes to step $k$, we first calculate the cumulative correlation $c$ between all the candidate regression variables and the current residuals. The aim of this is to find the regression variables that are most relevant to the current residuals. We define the maximum cumulative correlation at this time as $c_{max}^k$, and the set of candidate regressions that satisfy the maximum cumulative correlation is $A$. Thus,

$$c_{max}^k = \max_j \left\{ c_j^k \right\}, A = \left\{ j | c_j^k = c_{max}^k \right\} \tag{27}$$

We recombine all the regression variables in $A$ into a matrix $X_A$ of $n * |A|$ and calculate its least squares estimate:

$$\bar{Y}^{k+1} = X_A \left( X_A^T X_A \right)^{-1} X_A^T T \tag{28}$$

After calculating $\bar{Y}^{k+1}$, one can update $Y^k$:

$$Y^{k+1} = Y^k + \gamma^k \left( \bar{Y}^{k+1} - Y^k \right) \tag{29}$$

where $\gamma$ is the search step size of the $k$-th step. At the same time, according to the above formula, we can obtain:

$$X_A^T \left( \bar{Y}^{k+1} - Y^k \right) = X_A \left( T - Y^k \right) \tag{30}$$

Combining Formulas (27), (29) and (30), we can obtain:

$$c_j^{k+1}(\gamma) = |1 - \gamma| c_{\max}^k, \forall j \in A \tag{31}$$

$$c_j^{k+1}(\gamma) = \left| a_j^k - \gamma b_j^k \right|_1, \forall j \notin A \tag{32}$$

Among them, $a_j^k = \left(T - Y^k\right)^T x_j$, $b_j^k = \left(\bar{Y}^{k+1} - Y^k\right)^T x_j$. When there is a regressor $x_j$ of $j \notin A$ that makes the above two equations equal, this regressor will be added to the model in step $k + 1$, and the regression coefficient matrix will be updated as follows:

$$W^{k+1} = \left(1 - \gamma^k\right) W^k + \gamma^k \bar{W}^{k+1} \tag{33}$$

where $\bar{W}^{k+1}$ is a sparse matrix of $m * p$, its non-zero row order corresponds to $j \in A$, and the value of non-zero rows is determined by the row vector corresponding to $X_A \left(X_A^T X_A\right)^{-1} X_A^T T$ by the least square estimation. This can sort the regression variables.

### 4.3.2. Leave-One-Out Is Used to Determine the Optimal Number of Neurons to Retain

Since the MRSR only provides a ranking of the effects of each neuron on the output, it is still uncertain how many neurons are retained to minimize the error and achieve the best model effect. Therefore, LOO is used to make a decision regarding the actual optimal number of neurons in the model:

$$\varepsilon = \frac{y_i - h_i b_i}{1 - h_i P h_i^T} \tag{34}$$

where $\varepsilon$ represents the error, $P$ can be represented as $P = \left(H^T H\right)^{-1}$, and $H$ is the hidden layer output matrix of the ELM. $y_i$ represents the output after the $i$-th neuron is added, and $h_i$ and $b_i$ represent the weight and bias of the ELM input layer, respectively. After obtaining the order of the importance of neurons to the output, the number of neurons suitable for the model can be determined by evaluating the LOO error and the number of neurons used. We continuously add neurons to the model according to their order, test the error $\varepsilon$, and choose the lowest $\varepsilon$ as the number of neurons in the final model.

## 5. Experimental Results

The main purpose of the visual navigation in this paper is to obtain the best navigation performance for travelling from a random initial location to a goal location. In this section, we will train our model and evaluate the navigation results of our model against those of other baseline models on different tasks.

### 5.1. Experimental Simulation Environment

To effectively apply our model in different types of environments,, we use The House Of InteRactions (AI2THOR) framework proposed in [22]. It is a highly realistic indoor simulation environment built by the Allen Institute for Artificial Intelligence, Stanford University, Carnegie Mellon University, University of Washington and University of Southern California, in which agents can complete navigation tasks by interacting with the environment. AI2-THOR provides indoor 3D synthetic scenes, such as the room categories of living room, bedroom, and bathroom.

Parts of the scene are shown in Figure 5. In our experiments, the input includes two parts, $S = <F, T>$ and $F$, which represent first-person views of the agent, where the resolution of one image is $224 \times 224 \times 3$, and $T$ represents the goal. Each image is embedded as a 2048D feature vector, which is pretrained on ImageNet with Resnet-50. We consider four discrete actions: ahead, backwards, left, and right. The robot takes 0.5 m as a fixed step length, and its turning angle is 90 degrees in the environment.

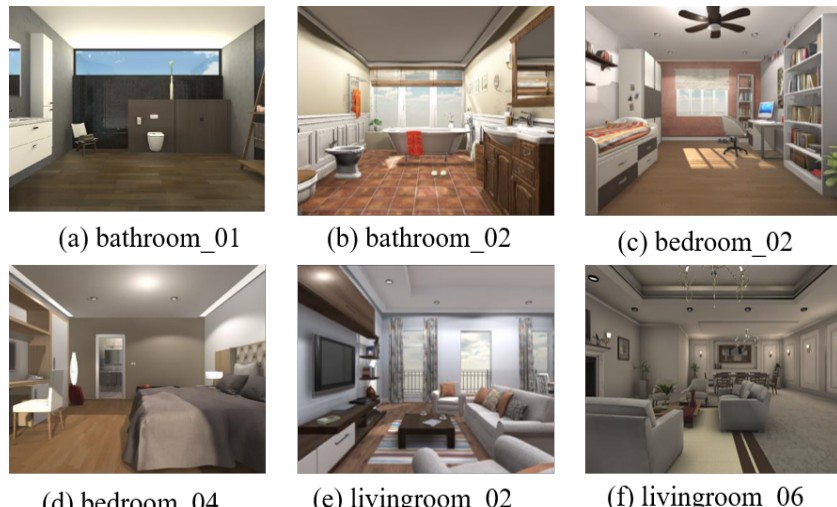

(a) bathroom_01　　　　(b) bathroom_02　　　　(c) bedroom_02

(d) bedroom_04　　　　(e) livingroom_02　　　　(f) livingroom_06

**Figure 5.** AI2-THOR scene. This is a good representation of the real world in terms of the physics of the environment.

### 5.2. Implementation Details

All of the models were implemented in the Ubuntu16.04 operation system and TensorFlow and trained/tested on an NVIDIA GeForce GTX 2070. For training, the navigation performance was assessed using 20 goals randomly chosen from indoor navigation environments in our dataset. All of the learning models were trained within 5 million frames, and it took about 1.5 h to pass one million training frames across all threads. The test ended when the robot either arrived at the goal location or after it took more than 5000 steps. For the purposes of our evaluation, we performed 1000 different tests for each scene.

In order to verify the effectiveness of our method more comprehensively, we compared the results obtained for the following three tasks:

(1) Task(I) : Navigation performance achieved using the trained target points in the trained scene. The trained model was used to verify the capability of a random starting point to the target point trained in the same scene.

(2) Task(II): Navigation performance of new target points in the training scene. The trained model was used to verify the capability of a random starting point to the untrained target point in the same scene.

(3) Task(III): Navigation performance achieved using similar target points in the new scene. The trained model was used to test the capability of a random starting point to the similar target point in the new scene. This helps to verify the generalization ability of the model in a new but similar scene to the training scene. The selected target points and the training target points belong to the same category, such as bookcases, washstands, switches, flower pots, etc.

### 5.3. Baselines

We compared our models with the following three baseline models:

(1) **Random**. At each time step, the agent randomly selects an action for navigation using a uniform distribution. Note that since the performance is extremely low for Task II and Task III, in this paper we only show the results obtained for Task I.

(2) **Reinforcement Learning (RL)**. In this paper, we utilize the method proposed in [22]. This baseline model updates its own strategy through constant interaction between the agent and the environment. Its reward function is set manually. This is the original target-driven method—it uses deep reinforcement learning and targets use and updates the same Siamese parameters but uses different scene-specific parameters

(3) **Inverse Reinforcement Learning(IRL)**. In this paper, we utilize the method proposed in [16]. The baseline architecture is similar to ours, but it requires knowledge of the expert trajectory to generate the reward.

### 5.4. Evaluation Metrics

In order to evaluate the performance of our proposed method, we applied three typical evaluation indicators—Success Rate (SR), Path Length Error (PLE), and Success weighted by Path Length (SPL) [23]—as defined by the following:

$$SR = \frac{N_s}{N_t} \tag{35}$$

$$PLE = L_t - L_b \tag{36}$$

$$SPL = \frac{1}{N_t} \sum_{i=1}^{N_t} S_i \frac{l_i}{\max{(p_i, l_i)}} \tag{37}$$

where $N_t$ and $N_s$ are the number of total tests and successful tests; $L_b$ and $L_t$ are the best path and actual path; $l_i$ and $p_i$ are the best path length and actual path length between the starting point and the target point; and $S_i$ is a binary indicator—if the agent navigates to the target point successfully, it will be recorded as 1, while otherwise, it will be recorded as 0.

Note that, in the metrics SR and SPL, higher values represent better performance; in contrast, lower values represent better performance in PLE.

### 5.5. Performance Comparison

In this subsection, we focus on four results: the efficiency during training, the performance of our approach in navigating to the trained targets, the new targets in trained scenes, and the new targets in new scenes. Additionally, the baselines are evaluated for comparison.

#### 5.5.1. Training Efficiency

As we know, training efficiency is important for deep learning-based methods. Figure 6 shows the error between the actual path length of the agent and the best path during training. Comparison results of training process. The black, blue, and red curves represent the training process of RL, IRL, and RELM-IRL (ours), respectively. It can be seen that our approach and IRL took less time to converge—both required about 3.5 million frames to learn a stable navigation policy, while it took nearly 4 million frames to train in RL.

Moreover, from the perspective of local details, our method maintained good performance after convergence, which is better than RL and IRL, indicating that our approach learns fast to obtain the best navigation policy. The reason for this might be that we used a regularization ELM method to reduce the complexity of the model, thereby, speeding up the training.

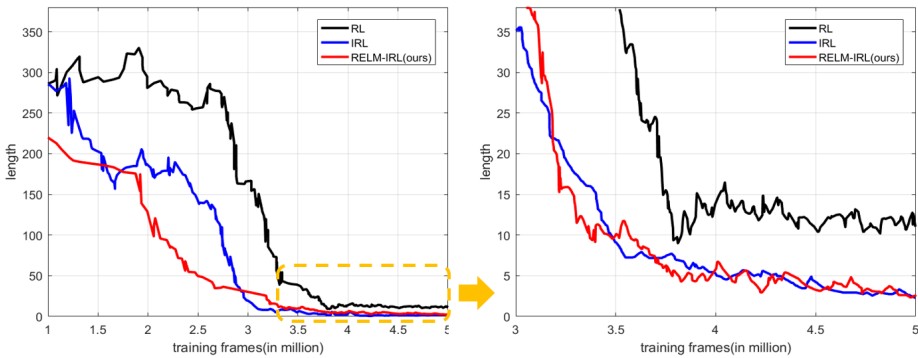

**Figure 6.** Comparison results of training process. The black, blue, and red curves represent the training process of RL, IRL, and RELM-IRL (ours), respectively.

### 5.5.2. Performance on Trained Targets (Task I)

To analyze the navigation performance of our approach, we first analyze the performance on the trained targets to reflect the navigation ability. The experimental results are shown in Table 1. These are the comparative experimental results of random agents, A3C based on RL, IRL, and our method. We focus on the performance of a random agent. In the bathroom, although the probability of the random agent successfully navigating to the target point is 20% in 1000 tests, its SPL value is very low, and the error with the best path length is large.

In the bedroom and living room scenes, as the scene becomes larger, the effect of the random agent drops sharply, making it difficult to accurately navigate to the target point. This proves that it is difficult to navigate through random selection in this simulation environment, and this also illustrates the validity of the experimental verification settings. In terms of the navigation success rate, it can be seen that when the model tends to be stable, A3C based on RL, IRL, and our method accurately navigated to the target point in each test regardless of whether this is in a small range of scenes or in a large range of scenes.

However, according to the results shown in Figure 5, it can be proven that our method can be stabilized faster. The navigation accuracy is also higher under the same number of training steps, which proves the effectiveness of our method. Combining PLE and SPL, our method performed better than IRL. This is consistent with the expected results, due to the simplified model resulting in a loss of information.

**Table 1.** Quantitative results on trained targets on Task (I). For SR and SPL, higher values represent a better performance. In contrast, lower values represent a better performance in PLE. The best results for individual scenes and the best average results across all scenes are shown in blue and red, respectively.

| Methods | Metrics | Bathroom | Bedroom | Livingroom | Ave. Performance |
|---|---|---|---|---|---|
| Random | PLE ($\downarrow$) | 83.27 | 82.86 | 79.09 | 81.74 |
| | SR ($\uparrow$) | 0.20 | 0.15 | 0.12 | 0.16 |
| | SPL ($\uparrow$) | 0.03 | 0.02 | 0.02 | 0.02 |
| RL | PLE ($\downarrow$) | 5.33 | 4.74 | 9.01 | 6.36 |
| | SR ($\uparrow$) | 1 | 1 | 1 | 1 |
| | SPL ($\uparrow$) | 0.56 | 0.68 | 0.64 | 0.63 |
| IRL | PLE ($\downarrow$) | 2.63 | 1.69 | 3.09 | 2.47 |
| | SR ($\uparrow$) | 1 | 1 | 1 | 1 |
| | SPL ($\uparrow$) | 0.76 | 0.74 | 0.82 | 0.77 |
| RELM-IRL (Ours) | PLE ($\downarrow$) | 2.53 | 1.21 | 2.38 | 2.04 |
| | SR ($\uparrow$) | 1 | 1 | 1 | 1 |
| | SPL ($\uparrow$) | 0.76 | 0.93 | 0.84 | 0.85 |

### 5.5.3. Generalization across New Targets in the Same Scenes (Task II)

To analyze the generalization performance of our approach, we first analyze the performance on the new targets in the same scenes (task II) to reflect the navigation ability. The experimental results are shown in Table 2. These are the experimental results of sampling the location around the training target point in several scenes during the training of the model or the results of using the position of the obvious object in the scene as the target point.

This sampling method takes into account that there are some extremely unclear target points in the scene, such as walls, glass, or other unique environments. At these target points, no matter which model is used, it is difficult to accurately navigate to the target point. It can be seen from the table that, compared with A3C and IRL, our method can greatly improve the navigation accuracy, shorten the average path length, and increase

the SPL value. For scenes, such as living rooms, the navigation effect is slightly reduced. However, overall the generalization performance of our method in training scenarios has been greatly improved. SR, PLE, and SPL have all improved. Compared with RL and IRL, the average SR navigation accuracy of our method is improved by 9% and 11%, respectively.

**Table 2.** Quantitative results for new targets in the same scenes in Task (II). The best results for individual scenes and the best average results across all scenes are shown in blue and red, respectively.

| Methods | Metrics | Bathroom | Bedroom | Livingroom | Ave. Performance |
|---|---|---|---|---|---|
| | PLE (↓) | 64.77 | 56.42 | 49.55 | 56.91 |
| RL | SR (↑) | 0.44 | 0.48 | 0.61 | 0.51 |
| | SPL (↑) | 0.16 | 0.23 | 0.29 | 0.23 |
| | PLE (↓) | 62.01 | 51.19 | 41.32 | 51.51 |
| IRL | SR (↑) | 0.49 | 0.41 | 0.57 | 0.49 |
| | SPL (↑) | 0.14 | 0.33 | 0.32 | 0.26 |
| | PLE (↓) | 52.43 | 56.04 | 41.03 | 49.83 |
| RELM-IRL (Ours) | SR (↑) | 0.61 | 0.51 | 0.69 | 0.60 |
| | SPL (↑) | 0.23 | 0.35 | 0.34 | 0.31 |

### 5.5.4. Generalization across Similar Targets in New Scences (Task III)

Moreover, in order to analyze the generalization performance of our approach, we additionally analyzed the performance on similar targets in the new scenes (task III) to reflect the navigation ability. This was conducted in order to verify the generalization ability of the model in a new but similar scene to the training scene. The selected target points and training target points belong to the same category, such as bookcases, washstands, switches, or flower pots, and the results are shown in Table 3.

Similar to the results for Task 2, this shows that the our method can improve the generalization ability to a certain extent by reducing the complexity of the model. Compared with RL, the average SR and SPL of our method are improved by 6% and 4%, respectively. For IRL, the average SR and SPL of our method are improved by 5% and 5%, respectively. The above results demonstrate the effectiveness and efficiency of our approach.

**Table 3.** Quantitative results on similar targets for the new scenes in Task (III). The best results in individual scenes and the best average results across all scenes are shown in blue and red, respectively.

| Methods | Metrics | Bathroom | Bedroom | Livingroom | Ave. Performance |
|---|---|---|---|---|---|
| | PLE (↓) | 82.01 | 83.94 | 77.96 | 81.30 |
| RL | SR (↑) | 0.23 | 0.09 | 0.08 | 0.13 |
| | SPL (↑) | 0.07 | 0.03 | 0.02 | 0.04 |
| | PLE (↓) | 78.81 | 80.44 | 83.45 | 80.90 |
| IRL | SR (↑) | 0.22 | 0.10 | 0.09 | 0.14 |
| | SPL (↑) | 0.06 | 0.02 | 0.02 | 0.03 |
| | PLE (↓) | 76.81 | 82.16 | 77.90 | 78.96 |
| RELM-IRL (Ours) | SR (↑) | 0.30 | 0.17 | 0.10 | 0.19 |
| | SPL (↑) | 0.09 | 0.09 | 0.05 | 0.08 |

### *5.6. Ablation Study*

In this section, we perform ablation on our method to gain further insight into our result. The results are as follows.

### 5.6.1. Effect of Different Numbers of Hiddden Layer Units in ELM

Before training the ELM, we first had to determine the number of hidden layer neurons. This number is related to the speed and accuracy of the ELM. Meanwhile, when the input and output dimensions are too high, it is difficult to use a small amount of hidden neuron layer units to obtain better results; thus, the number of hidden layer units was set from low to high. As shown in Figure 7, the number of hidden layer neurons was set from 1 to 8000. It can be seen that as the number of ELM hidden neurons increases, the test accuracy of the ELM gradually improves. When the number of neurons exceeds 1800, the accuracy fluctuates between 0.9 and 1. However, when the number of neurons exceeds 4700, the model will have large errors, and its performance will drop sharply.

Considering the complexity and computing speed obtained for the model in training, we chose to use 1950 hidden layer neurons for the experiment. The experimental results are shown in Figure 8. The blue data points are the predicted values of the model, and the red points are the real data of the test set itself. It can be seen that the difference between the prediction of the ELM and the real data is small, which shows that after sampling training ELM can output more accurate reward function parameters according to the image, indicating the effectiveness of ELM-IRL.

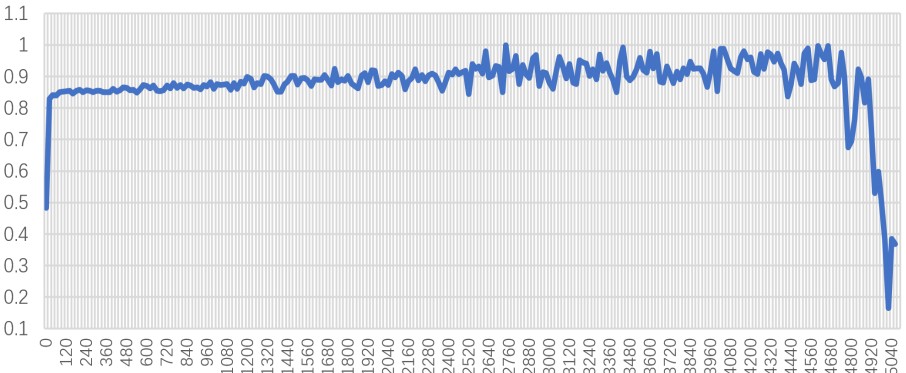

**Figure 7.** The encoded accuracy with different numbers of hidden layer units in ELM. It can be seen that the accuracy gradually improves up to the level of 1800 hidden neurons. However, when the number of neurons exceeds 4700, the model will have large errors and its performance will drop sharply.

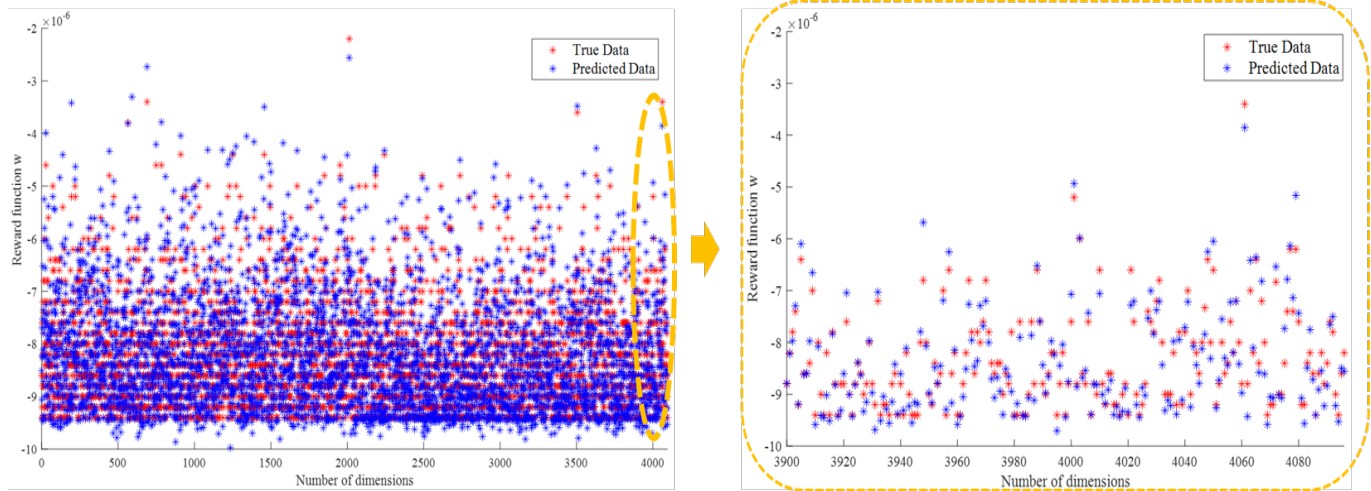

**Figure 8.** Test results of ELM under fixed neurons. The blue and red dots represent the output of ELM and the true data. It can be seen that the encoded error of ELM is very small, which verifies the effectiveness of ELM.

### 5.6.2. Effect of Regularization in ELM

In the regularized ELM experiment, the hidden layer neurons are sorted by the MRSR, the sorted neurons are gradually added using LOO, and the loss function is calculated to determine the optimal number of hidden layer neurons. Since the previous regularization method MRSR only provides a ranking of the effects of each neuron on the output, it is still uncertain how many neurons should we use to achieve the good model effect under limited neurons. Therefore, LOO is used to make a decision regarding the actual optimal number of neurons in the model. Once the importance of neurons to the output has been sorted, LOO can determine the number of neurons to use according to the loss of training, continuously add neurons to the model according to the order, test the error, and choose the smallest error for the number of neurons in the final model. The experimental results are shown in Figure 9. It can be seen that the LOO loss reaches the minimum when the number of hidden layers is 593; thus, the ELM was set to take the first 593 hidden layer units of the multi-response sparse regression to form the hidden layer.

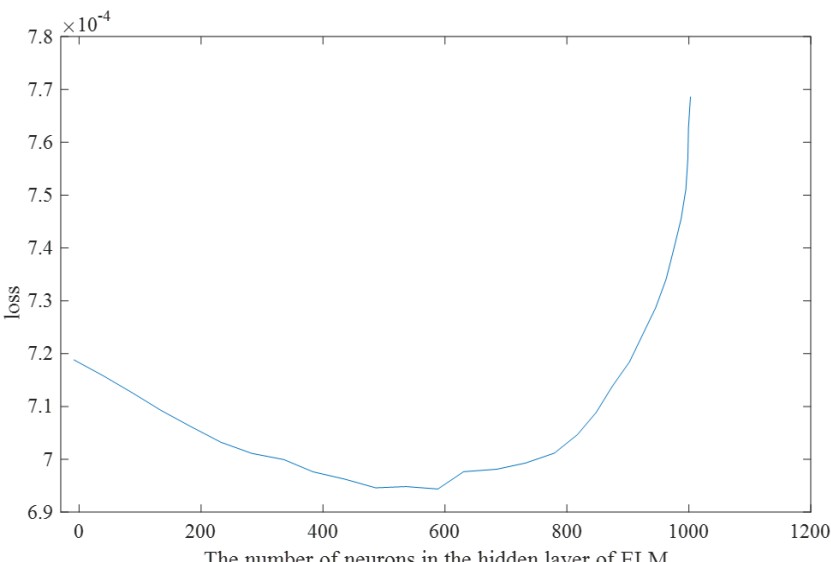

**Figure 9.** Relationship between the LOO calculation loss function and the number of neurons. It can be seen the best choice for the number of neurons is in the middle at about 593 hidden layer neurons.

Figure 10 shows the test results of the ELM with 593 hidden layer neurons. The blue data points in the figure are the predictions of the model, while the red points are the true data of the test set. The gap between the prediction of the ELM and the real data obtained for the sample is extremely small. Compared with the model without regularization, the prediction error is further reduced, and the change trend and amplitude are basically the same, which proves the effectiveness of the regularized ELM.

### 5.6.3. Navigation Performance Using ELM-IRL and RELM-IRL

In order to illustrate the generalization performance of the regularized ELM-IRL, we conducted an experiment under task III and selected SPL as the evaluation metric. The results are shown in Figure 11. It can be seen that the performance of the ELM-IRL after regularization is better than that of ELM-IRL, which proves that the generalization ability of the model was effectively improved after regularization.

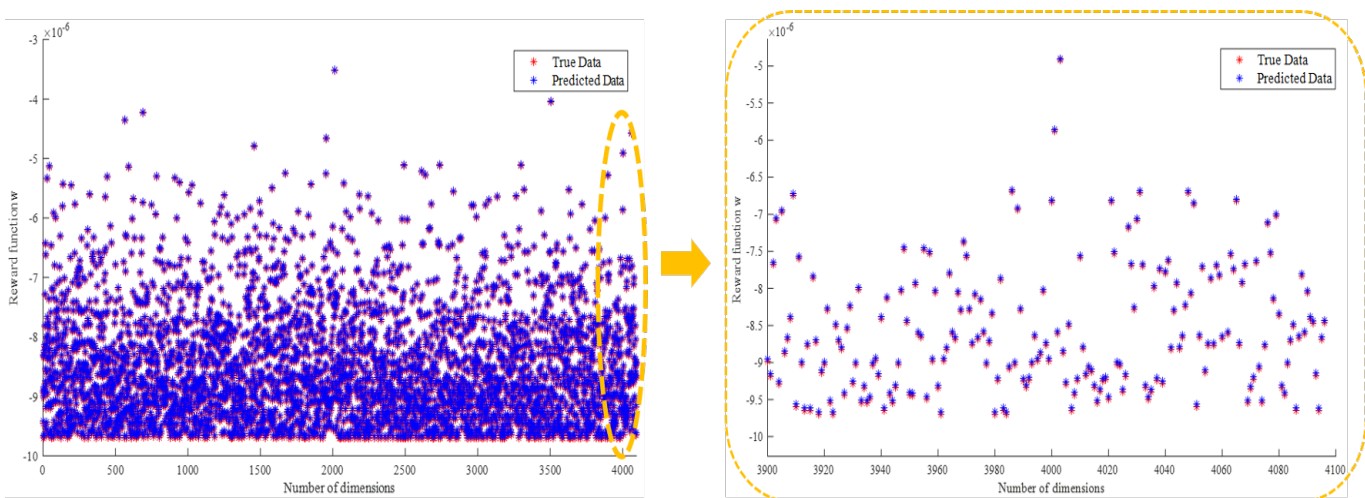

**Figure 10.** Result after ELM regularization. Blue data points are the output of the regularized ELM and the red points are the true data.

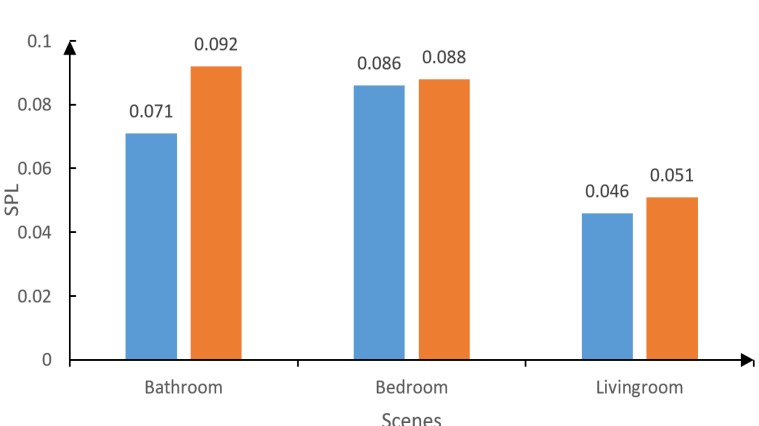

**Figure 11.** Experimental results of regularized ELM. The orange and blue represent the SPL of ELM-IRL with and without regularization, respectively.

## 6. Conclusions

This paper studies the navigation problems of mobile robots in indoor scenes. Our proposed method is different from previous navigation methods that use SLAM, a priori maps, etc. Based on the RL methods that have been widely used in recent years, the purpose of this paper is to train the intelligence of the agent using information obtained by visual sensors. We propose a regularized ELM-IRL navigation method to realize the end-to-end navigation of mobile robots in indoor scenes. The experimental results show that our method has a higher SR and SPL in many tasks compared with previous end-to-end approaches, thus, demonstrating the effectiveness and efficiency of our approach.

**Author Contributions:** Conceptualization, Q.F. and W.Z.; methodology, Q.F.; software, W.Z. and X.W.; validation, W.Z. and X.W.; formal analysis, W.Z.; investigation, Q.F. and X.W.; resources, Q.F. and X.W.; data curation, W.Z. and X.W.; writing—original draft preparation, W.Z. and Q.F.; writing—review and editing, Q.F. and W.Z.; visualization, W.Z.; supervision, Q.F.; project administration, Q.F.; funding acquisition, Q.F. All authors have read and agreed to the published version of the manuscript.

**Funding:** This work was supported by the National Natural Science Foundation of China (Grant Nos. 61703418 and 61825305).

**Conflicts of Interest:** The authors declare no conflict of interest.

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
