# Peer review of "Visual Navigation Using Inverse Reinforcement Learning and an Extreme Learning Machine"

_electronics, doi:10.3390/electronics10161997_

Round 1

Reviewer 1 Report

This paper presents a novel method for training autonomous platforms for navigation using the combination of  SOTA  ML/AI  methods  - Inverse reinforcement learning and extreme learning machine. 

The proposed method is sound and valid. Authors provide good argumentation for the  introduction of each part of the proposed algorithm. IRL is used because the reward function is hard to define analytically, so it must be learned from the data. ELM is simpler and easier to train than deep learning architectures. Regularization is introduced to obtain more general results. 

The novelty and contributions of the paper  are:

-a novel framework for navigation utilizing IRL and ELM

-introduction of regularization for achieving result that generalize better

-rigorous evaluation of the obtained navigation system in comparison to SOTA methods.

The paper is well structured and mostly easy to follow.

Some remarks

-english  should be proofread and some sentences should be rephrased with the help of native english speaker

Equation (3) introduces summation over i=1 to d … what is d?

line 176 -  acronym A3C is not explained (not in acronym list) 

line 262 - fig 3 or fig 4, please check

section 4.3.1 - check if all labels are explained. Also, how this part relates to the task of feature selection

“...Combining formulas (17), (19), (20), we can get:...”  update the formulas labels

 Section 5.3. please cite the source,  (if equation for SR, PLE and SPL are not first introduced in this paper) 

Figure 6 should be improved (use more contrast colors than red and orange, add legend, explain transparent lines in the background)

Author Response

Dear  Reviewer:

     Thank you for allowing a Minor Revisions of our manuscript, with an opportunity to address the reviewers’ comments. Those comments are all valuable and very helpful for revising and improving our paper, as well as the important guiding significance to our researches. We have studied comments carefully and have made corrections, and our point-by-point responses to the comments (Please see the attachment), which we hope meet with approval.

    Best regards,

< Qiang Fang > et al.

Reviewer 2 Report

Point 1 : Please explain the omitted abbreviations.

Point 2 : Please clearly express the relationship between A3C and fully connected layer in Figure 1.

Point 3 : Please indicate the relationships in the order of progression in Figure 3

Point 4 : Please indicate what the x-axis and y-axis represent in Figure 5, Figure 7, Figure 8, Figure 9, and Figure 10. And I want you to indicate what each color represents on the graph.

Figure 5, Please add a detailed description of the A12-THOR database in section 5.1.

Point 6 : In section 5.5.2, please provide a detailed explanation of why you chose and used the LOO method.

Author Response

(The authors gave the same response as above.)
